# Effects of Cannabis Legalization on Road Safety: A Literature Review

**DOI:** 10.3390/ijerph20054655

**Published:** 2023-03-06

**Authors:** Francisco González-Sala, Macarena Tortosa-Pérez, María Peñaranda-Ortega, Francisco Tortosa

**Affiliations:** 1Departamento de Psicología Evolutiva y de la Educación, Universidad de Valencia, 46010 Valencia, Spain; 2Departamento de Psicología, Universidad Internacional de Valencia-VIU, 46002 Valencia, Spain; 3Departamento de Psicología Básica y Metodología, Universidad de Murcia, 30100 Murcia, Spain; 4Instituto de Investigación en Tráfico y Seguridad Vial-INTRAS, Universidad de Valencia, 46010 Valencia, Spain

**Keywords:** cannabis, marijuana, legalization, traffic accidents, road safety, systematic review

## Abstract

Background: Legalizing medical and recreational cannabis and decriminalizing this substance may have unanticipated effects on traffic safety. The present study aimed to assess the impact of cannabis legalization on traffic accidents. Methods: A systematic review was carried out following the Preferred Reporting Items for Systematic Reviews and Meta-Analyses (PRISMA) declaration of the articles included in the Web of Science (WoS) and Scopus databases. The number of papers included in the review was 29. Results: The results show that in 15 papers, there is a relationship between the legalization of medical and/or recreational cannabis and the number of traffic accidents, while in 5 papers, no such relationship is observed. In addition, nine articles indicate a greater number of risk behaviors related to driving after consumption, identifying young, male, and alcohol consumption together with cannabis as the risk profile. Conclusions: It can be concluded that the legalization of medical and/or recreational cannabis has negative effects on road safety when considering the number of jobs that affect the number of fatalities.

## 1. Introduction

Drug consumption is a market that generates high profits. Although cannabis is not the illegal substance that generates the most money due to its low cost, it is the most consumed, abused, and trafficked substance [1]. Currently, a series of policies are being adopted in different countries in favor of the legalization of this substance, whether medicinal or recreational, as well as decriminalization campaigns in relation to its use.

Colorado and Washington were the first US states to legalize this substance for recreational use in 2012. In 2013, Uruguay became the first country in the world to legalize the sale and cultivation of recreational marijuana. Canada was next, adopting the measure in 2018, being the first G7 country to do so [2,3].

With all these changes, different studies have begun to collect scientific evidence on the effects that the implementation of this measure can produce on various aspects of public life, such as health [4], and the prevalence of consumption or its relationship with road safety and traffic [5,6,7], among others.

Regarding traffic and driving, Hartman and Huestis [5] point out that recent THC consumption of between 2 to 5 nG/mL in the blood is associated with the inability to drive a vehicle. Along the same lines, Li et al. [6] conducted an extensive meta-analysis in which they concluded that cannabis use by drivers is associated with a significant increase in the risk of being involved in a traffic accident. Similarly, Woo et al. [8] indicate that cannabis use predicts risky behavior in fatal traffic crashes, despite alcohol remaining a better predictor. Regarding the consequences brought by the legalization of recreational cannabis, according to Roffman [7], the number of drivers driving under the influence of cannabis increased after its legalization in Washington, from 22.2% of cases to 32.7%. Likewise, they also observed a 111% increase in the number of fatal traffic accidents where cannabis was involved in 2014 versus the 4 years prior to legalization. Kamer et al. [9] report an increase in fatal traffic accidents following the legalization of recreational cannabis in the states of Alaska, Colorado, Oregon, and Washington compared to 20 states that had not legalized either medical or recreational cannabis.

In Canada, according to the National Cannabis Survey [10], there has been no detectable increase in drivers who have consumed in the previous two hours, before and after legalization, but there has been an increase in cases of drivers who tested positive for cannabis, of which 20% also report having consumed alcohol, after legalization.

Other studies contradict these results. Romano et al. [11] do not find a significant relationship between testing positive for cannabis as a driver and the risk of having an accident, but there is a relationship with other illegal drugs, mainly alcohol [12]. For their part, Fergusson and Horwood [13] attribute the association between cannabis use and the risk of having a traffic accident to the characteristics of young people, who tend to use cannabis, rather than to the effects of this substance on driving ability, since cannabis users have a greater tendency to drive under the influence of alcohol and engage in risky driving behaviors, as well as a greater tendency to violate traffic regulations [14,15,16].

These contradictory results only increase the uncertainty regarding the actual effect of cannabis legalization on trafficking, which makes the search for scientific evidence even more necessary. In this regard, Palamar et al. [17] point out that with legalization, marijuana consumption may increase due to an increase in the acceptability of this substance and a decrease in the perception of its danger, causing, in turn, a significant increase in the number of individuals who drive under the influence of cannabis. Regarding causality, Cavazos-Rehg et al. [18] argue that the risk of an accident is not a direct cause of legalization, since the decision to drive under the influence of cannabis is a purely individual act.

The aim of the present study was to determine the effect of the legalization or decriminalization of medical and/or recreational cannabis on fatal and non-fatal traffic accidents by means of a systematic review, providing arguments based on scientific evidence to help in decision-making and the design of preventive programs. The following hypotheses are proposed:

**Hypothesis** **1.**
*There is a positive relationship between the legalization of medical cannabis and the incidence of traffic accidents.*


**Hypothesis** **2.**
*A positive relationship exists between the legalization of recreational cannabis and the incidence of traffic accidents.*


**Hypothesis** **3.**
*There is a positive relationship between cannabis legalization and a higher number of risky behaviors associated with driving, such as lower use of protective driving devices, reckless driving, and riding as a passenger with a driver who has consumed before driving.*


**Hypothesis** **4.**
*Drivers who test positive for cannabis while driving have a higher risk profile, such as being male and young.*


## 2. Materials and Methods

### 2.1. Search Strategy

This systematic review was conducted following the Preferred Reporting Items for Systematic Reviews and Meta-Analyses (PRISMA) statement [19] (for further details of the PRISMA process, see Appendix A). The search was performed on 4 August 2022, in the Web of Science Core Collection (WOS) database and Scopus. The keywords selected in the subject field were (“driver* OR motor vehicle* OR road safety* OR car accident* OR crash fatalit* OR driving risk* OR injured driver* OR vehicle*”) AND (“legalization*”) AND (“marijuana*” OR cannabis*”). The search period covered the last 10 years, from 2012 to 2021.

### 2.2. Selection Criteria

The inclusion criteria were scientific articles written in English or Spanish that compared results on traffic accidents and marijuana use before and after legalization or between states or countries that have legalized recreational and/or medical marijuana and those that have not.

As exclusion criteria, other documentary typologies were established, such as reviews, letters to the editor, conference proceedings, or book chapters written in languages other than English and Spanish, or that did not compare pre- and post-legalization periods or different states.

### 2.3. Study Selection

The number of papers found in WoS was 103, and in Scopus 86 (see Figure 1). After reviewing the references of the selected articles in a reverse search process [20] in order to identify useful references that had not appeared in the initial search, 8 more articles were included. The total number of articles reviewed was 197, of which 68 were eliminated due to duplicates. Once a review of the title and abstract was performed, 78 papers were eliminated, mainly because they did not include a comparison between countries or states that had or had not legalized the use of cannabis, or because they did not compare the periods before and after legalization, and one study for being a conference paper. After downloading and reading the 49 selected articles, 20 were eliminated because they did not address the effects of cannabis legalization in relation to road safety, bringing the total number of articles included in this review to 29 (see Figure 1).

This entire process of analysis to determine the suitability or otherwise of the articles to the research objective, as well as the extraction of the results according to the variables under study, was carried out independently by two researchers who acted as “blinded” evaluators. In cases in which there was disagreement, mainly over some results related to the objectives of the systematic review, a third researcher participated in the decision-making, and the results were accepted as valid by consensus after addressing the discrepancies between evaluators.

The methodological quality of the articles selected for the present systematic review was assessed through 10 of the 18 indicators included in the SQUIRE Guidelines 2.0 quality scale [21]. These indicators were title, abstract, problem description, available knowledge, specific aims, measures, analysis, results, limitations, and conclusions. The articles were classified into three categories according to their quality—low, medium, and high. This process was carried out by two of the researchers, who concluded that the articles included in the review were in the high category, so none of the selected articles were eliminated.

## 3. Results

### 3.1. Characteristics of the Studies Included in the Systematic Review

Of the 29 articles included in this review, eight focused on the legalization of medical cannabis [22,23,24,25,26,27,28,29], with respect to recreational cannabis, there were 13 papers that addressed the impact of its legalization with respect to traffic accidents [3,30,31,32,33,34,35,36,37,38,39,40,41], two papers included both medical and recreational cannabis [42,43], and seven papers discussed the decriminalization and legalization of cannabis in general [8,43,44,45,46,47,48].

Regarding the origin of the samples, two papers are from Canada [23,35], one paper is from Uruguay [34], one paper includes samples from three countries—the United States, Canada, and England [23]—and the remaining papers, 25 in total, are from the United States, with the states of California, Colorado, and Washington being the most represented. With regard to the samples, it should be noted that these were mainly made up of men, except for two studies with a greater presence of women [3,23], with a minimum age of 14 years [26,48]. Among the selected papers were a large number of studies that, in addition to collecting data on cannabis use, also reported on alcohol use [3,8,22,23,24,25,28,29,36,37,44,45,46].

Of the 29 articles included in the present review, 16 articles were collected from the Fatality Analysis Reporting System (FARS) database [8,24,25,26,27,28,29,31,32,37,38,39,40,43,46,47]. This database collected nationwide information on about 100 indicators related to the driver, the vehicle, and the circumstances associated with the crash in cases where a driver or non-driver death occurred within 30 days of the crash. The relevance of this database for studies addressing the effect of cannabis legalization on road safety was due to the fact that it included tests for different drugs, including the presence of cannabis in blood and/or urine. However, it also had limitations, as reflected in the analyzed studies, such as the fact that drug tests were not always performed, or the existing differences between states in the regulation of the level of THC in blood when testing positive, an aspect that had been mentioned as a limitation in different studies [24,26,27,31,43,46,47].

### 3.2. Relationship between the Legalization of Medical Cannabis and Traffic Accidents

Table 1 and Table 2 list the studies that have addressed the impact of medical cannabis legalization in relation to traffic accidents. Three studies compared the effect of legalization at two time points, before and after legalization [22,24,28], five studies compared states that have legalized medical cannabis with states that have not [25,26,27,28,29], and one study compared countries with legalization laws, the United States and Canada, with England, a country that has not legalized cannabis [23].

As shown in Table 2, two papers found increases in the number of fatal accidents after legalization [22,24]. Jones et al. [22] report a downward trend in the number of drivers testing positive for cannabis in the years prior to legalization and in the years after legalization if legalization had not been approved. After legalization, the trend reversed, with an increase in the number of cases, although this change was not significant. Steinemann et al. [36] analyzed traffic crashes occurring pre- and post-MCL in Hawaii, concluding that crashes resulting in death nearly tripled from 5.5% between 1993 and 2000 to 16.3% between 2011 and 2015. However, they found no change in the prevalence of alcohol or other substances, such as methamphetamine. Drivers who tested positive for THC in drug tests were usually younger; the accident typically occurred at night and fewer protective mechanisms, such as seat belts, were used. In addition, in 63% of cases, the dual use of alcohol and cannabis was observed.

When comparing states that legalized medical cannabis and those that had not, Solomonsen-Sautel et al. [28] detected a positive trend in the proportion of fatal traffic crashes involving cannabis in Colorado compared to those states that had not legalized it; however, this trend was not observed in the proportion of fatal traffic crashes involving alcohol consumption. More specifically, Colorado saw an increase in drivers testing positive for cannabis between 1994 and 2011, decreasing in the pre-legalization period (−0.19 (0.08) *p* = 0.0227), and increasing again significantly after legalization (2.16 (0.45) *p* = 0.0001). This trend was significantly different from that in states that have not legalized medical cannabis use, which remained stable over the years.

On the other hand, other papers found mixed results, indicating increases only in some states [25,26,27], while a single paper found no relationship between legalization and fatal traffic accidents [29]. Sevigny et al. [25] found that the MCL did not have an effect on the number of drivers under the impact of cannabis, except in the case where the sale through dispensaries had been regulated, where he did find a significant increase of 14% in the number of drivers using cannabis in 2014, that is to say, a rise from 87 to 113 drivers testing positive in this type of accident per year. In addition, among illegal drugs, it is cannabis that was found to be most prevalent in drivers (8.8%) compared to stimulants (6.1%) or narcotics (4.2%), among others. In the case of alcohol, the percentages were around 25%. The highest percentage of cannabis-positive drivers involved in fatal accidents was in states with marijuana decriminalization policies (28%), 16% in states with THC laws, 14% in states with medical marijuana laws, and just 0.2% in states with recreational marijuana laws.

Santaella-Tenorio et al. [26] found a decrease in fatal accidents after legalization, although this decrease only occurred in 7 states and in drivers between 15 and 44 years of age.

Masten and Guenzburger [27] detected an increase in the number of drivers under the impact of cannabis in the period from 1992 to 2009; this study employed an interrupted series design analyzing twelve states (Alaska, California, Colorado, Hawaii, Michigan, Montana, Nevada, New Mexico, Oregon, Rhode Island, Vermont, and Washington) in which medical marijuana had already been legalized. Still, they only found a significant increasing trend in the state of Washington, with an increase of 3.4 percentage points in the number of cases and a 4.6-percentage point increase in injured drivers; Hawaii, with a 6-percentage point increase in the number of drivers and a 9.6-percentage point increase in drivers with injuries; and California with a 2.1-percentage point increase in the number of drivers and a 5.7-percentage point increase in drivers with injuries.

Finally, Anderson et al. [29] found no significant differences in the increase in fatal accidents after the MCL in the 19 states that took this measure but a decrease in the number of fatal accidents where alcohol was present, suggesting that cannabis consumption exerts a substitutive effect to alcohol consumption. The study also showed a significant decrease of between 10% and 13% in the mortality rate in the first 4 years after legalization, although this effect was no longer significant in the fifth year.

In a comparative analysis between the United States, Canada, and England conducted by Wadsworth and Hammond [23], the authors observed higher cannabis use, greater access to cannabis, lower risk perception, and a higher prevalence of cannabis-impaired drivers in the first two countries compared to England; this is, according to the authors, due to the prohibitive laws in England, as well as to previous trends already existing in relation to consumption. Specifically, 15.4% in Canada, 9.4% in England, and 27.7% in the United States had driven within two hours after consuming cannabis; 18%, 11%, and 25%, respectively, had been passengers in a car in which the driver had consumed; and with regard to the perceived risk of having an accident after consumption, 53.2% in Canada, 56.7% in England, and 43.6% in the United States considered this risk to be high.

### 3.3. Relationship between the Legalization of Recreational Cannabis and Traffic Accidents

Table 3 and Table 4 list the studies that have addressed the impact of recreational cannabis legalization in relation to traffic accidents. Eleven studies have compared the effect of legalization at two time points, before and after legalization [3,30,31,32,34,35,36,38,39,40,41], and six studies have compared states that have legalized recreational cannabis versus states that have not [3,33,37,39,40,41].

In relation to the legalization of recreational cannabis, seven studies have found an increase in traffic accidents in some of the states that legalized recreational cannabis after its legalization [3,31,32,34,36,38,41]. One study indicates increases after the opening of dispensaries [40], while two studies found no effect on road safety [30,39]. Meanwhile, of the studies that compared states that have legalized cannabis with those that have not, four found a greater effect in states that have legalized recreational cannabis [3,33,40,41], and two studies find no significant effect [37,39].

Focusing in more detail on the studies included in this section (Table 3 and Table 4), Delling et al. [3] focus on whether, after MRL in Colorado, there has been an increase in traffic accidents using Oklahoma and New York as control states, with an increase in fatal accidents in Colorado compared to control states after legalization. In addition, there has been a significant increase in hospitalizations for cannabis use in Colorado and alcohol use compared to the control states.

Hansen et al. [37] sought to test whether MRL has increased fatal traffic accidents involving marijuana or alcohol. Their results indicate that there is no such increase in the states of Colorado and Washington. In their analyses, they try to look for a causal effect between these two factors by observing similar changes related to marijuana, alcohol, and the number of traffic accidents in other states despite the fact that recreational marijuana has not been legalized in these control groups.

Aydelotte et al. [39] sought to analyze the effect of MRL on fatal traffic accidents in Colorado and Washington between 2009 and 2015 compared to other states that have not legalized its use. The study reveals that no significant differences are found in the number of fatal crashes before legalization with respect to control states. In the three years following MRL in Colorado and Washington, there was a slight increase in the total number of fatal traffic accidents, but this is not significant compared to other control states.

Aydelotte et al. [40] analyzed changes in Colorado and Washington following RCL versus states that had MCLs (Hawaii, Montana, New Mexico, Rhode Island, and Vermont) or no legalization laws (Idaho, Kansas, Nebraska, and South Dakota), finding a non-significant increase in fatal traffic crashes over what was expected in the five years following legalization versus control states. However, there was a significant increase in fatal crashes once commercial recreational marijuana dispensaries began to open rather than when the legislative measure was implemented.

Lane and Hall [41] use an interrupted series analysis to analyze the cases of Colorado, where they observed an increase in the rate of fatal traffic accidents after legalization; Washington, where they observed an immediate increase followed by a decrease; and Oregon, where they found no significant difference from the previous trend. The study concludes that after MCL, there is an increase of one fatal crash per million population in the RCL and surrounding states.

Santaella-Tenorio et al. [38] focused on whether RCL is associated with the number of traffic fatalities in Colorado and Washington between 2005 to 2017, finding an increase in mortality in Colorado since 2014, in which recreational marijuana was legalized, specifically 75 more deaths per year. By contrast, in Washington, no such increase is observed after legalization. Moreover, in the United States, specifically in California, Borst et al. [36] analyzed the presence of cannabis and other drugs in traffic accidents in San Diego. The results indicate an increase over the years of cases of drivers presenting injuries after suffering a traffic accident, from 7.3% in 2010 to 14.8% in 2018. In addition, those in the THC+ group tend to be younger (X = 27 years) than those who do not use any drugs (X = 38 years), are more likely to be involved in a motorcycle accident, and less likely to use protective mechanisms such as helmet and seat belt, differences that proved to be statistically significant between the two groups. Finally, in fatal crashes in which drug testing was performed, in 27% of cases, the driver tested positive for THC.

Windle et al. [32] analyzed fatal traffic accidents and traffic fatalities in 10 states and the District of Columbia. The results point to a 15% increase in fatal traffic accidents after legalization and a 16% increase in traffic fatalities, leading to 308 more deaths each year after RCL.

Tefft and Arnold [31] focused on Washington State drivers involved in fatal traffic crashes to determine whether THC was present in blood and urine before and after the MRL. Specifically, 735 fatal crashes where the driver tested positive for THC in blood or urine, 3528 crashes where the driver tested negative for THC, and 4019 cases where drug tests were either not performed or the drug test results were not subsequently confirmed were analyzed. The results of the study indicate an increase of almost 10 points in the number of drivers who tested positive for THC after legalization, from 9.3% to 19.1%, and increasing THC levels in the blood. At the same time, a significant increase in the use of other drugs was detected in drivers who tested positive for THC after legalization.

Lensch et al. [33] mainly analyzed the attitudes and behaviors associated with driving after cannabis use in states that have legalized recreational marijuana and have legal marijuana dispensaries versus states that, despite having legalized it, do not have dispensaries, and whether the frequency of use may interfere with this type of behavior. The results of the study indicate that the use of cannabis while driving in the last 30 days is higher in states where marijuana has been legalized (7.3%) than in those where it has not (5.5%), and there is also a higher prevalence of traveling with a driver who has used (10.1% vs. 8.4%). However, protective driving behaviors also increase, mainly in states where cannabis has been legalized; there is a greater awareness of the danger of driving after having consumed, 52.1% compared to 39.7% in states that have not legalized it, as well as believing that the probability of having an accident after having consumed increases, 57.1% compared to 52.2%. All of these results were statistically significant. After stratifying the data by frequency of use, in states where recreational marijuana has been legalized, there is a significant decrease in risky driving compared to states where it has not.

Meanwhile, Rotermann [35] analyzed the changes in Canada after legalization with respect to the consumption habit, the origin of the cannabis, driving after having consumed, and riding with a driver who had consumed the drug. With respect to these last two variables, the results of the study indicate that there is no significant increase in driving within two hours of having consumed, this behavior being more frequent among men than among women, and among those who report higher consumption on a regular basis. In addition, there was a decrease from 5.3% to 4.2% in being a passenger in a vehicle the driver of which had consumed. This decrease occurs among women over 25 years of age and in Newfoundland and Labrador, Ontario, and Alberta territories, and does not occur in the rest of the country. This type of behavior occurs mainly in males, between 18 and 24 years of age, and in cases where there is consumption by both the driver and the passenger.

Callaghan et al. [30] evaluated whether there was an increase in the number of injuries after a traffic accident before and after the legalization of recreational marijuana. The results indicate no increase in driving-associated injuries after cannabis legalization among young drivers, with a slight decrease of 0.66 medical visits after legalization in Alberta and an increase of 0.09 visits in the case of Ontario. There are no differences in Alberta for all drivers, young or not, although there is an increase of 9.17 points in the number of medical visits after legalization (*p* = 0.52). These differences do exist in Ontario, with an increase of 28.93 visits (*p* = 0.30). In the specific case of this study, the researchers point out the role that variables such as the period of decriminalization prior to regulation, a postal service strike, or a higher penalty for driving under the influence of cannabis and cannabis and alcohol after legalization may have played in the results.

In the case of Uruguay, Nazif-Muñoz et al. [34] evaluated mortality rates in traffic accidents after MRL in urban and rural areas. The results of the study indicate a 52.4% increase in the mortality rate of drivers after legalization. However, this increase was not significant in the case of motorcyclists, and there was a significant increase in the number of traffic accidents after legalization. In addition, the study points out how this increase is significant in the country’s capital, Montevideo, as opposed to rural provinces.

### 3.4. Legalization and Decriminalization of Medical and Recreational Cannabis

Only two studies reflect the joint impact of medical and recreational cannabis on road safety [42,43] and seven studies address the impact of cannabis discrimination [8,43,44,45,46,47,48]. Regarding legalization (Table 5 and Table 6), medical cannabis has a greater effect on the number of fatal accidents than the legalization of recreational cannabis [42], while [43] reports a greater effect of the legalization of recreational cannabis. Decriminalization seems to have an effect on road safety [8,43,46,47,48], while other studies find no relationship with the number of traffic accidents [44], or with the number of traumatic injuries [45].

Focusing on studies that have addressed the legalization of cannabis in general or its decriminalization (Table 5 and Table 6), the work by Benedetti et al. [42] aimed to find out whether MCL and RCL are related to a higher number of drivers over 18 years of age who have driven under the influence of marijuana. The results of the study indicate that in states where medical marijuana has been legalized, more drivers report driving under the influence of marijuana than in states where neither medical nor recreational marijuana has been legalized, with a higher probability of 29%, and in states where both forms are legalized. In addition, the study picks up the role that the existence of laws controlling the THC threshold may have since, in the states where this threshold is regulated, the probability of driving after consuming marijuana is significantly reduced by 26% compared to states that do not have such laws.

Lee et al. [43] analyzed the number of accidents resulting in death according to legislative changes related to cannabis, observing that there were no significant differences in Arizona and New Jersey compared to states that acted as a control group, there being even a decrease, although this is not significant. By contrast, in Massachusetts, the numbers increase by 174.5%, in Connecticut by 75.3%, in Washington by 31.2%, and in Colorado (excluding Denver) by 63.1%. The authors conclude that MCL does not imply an increase in fatal crashes where the driver tests positive for THC. However, when these changes affect the decriminalization and legalization of recreational marijuana, there is a significant increase in the number of THC-positive cases among drivers in fatal crashes in both Washington and Colorado.

Couper and Peterson [48] evaluated the possible effect of marijuana legalization on the prevalence of this substance in the blood of drivers suspected of having consumed and showing some inability to drive, noting that after legalization, the number of cases of drivers who tested positive for THC increases from 4809 cases in 2009 to 5468 cases in 2013. The study points out a stabilization in the number of cases between the years 2009 and 2012, when marijuana was not yet legalized, while in 2013, when it was legalized, the number of cases increased considerably, with an increase from 5.8% to 12.1%. These drivers were mostly men, with percentages between 77% and 80% depending on the year of study, with the group between 21 and 30 years of age being the most represented, followed by the group under 21 years of age.

Woo et al. [8] focused on the relationship between THC and risky driving behaviors such as speeding or driver errors such as lane changes, and reckless driving, among others, which may lead to a fatal accident involving the driver, passenger, or pedestrian. The results of the study indicate that being young, male, driving a motorcycle, testing positive for alcohol in breathalyzer tests, delta 9-THC or carboxy-THC and other drugs are considered risk factors in traffic accidents involving high-speed driving. Carboxy-THC significantly predicted speeding (54% more likely than those who did not test positive) and the presence of driving errors. Positive delta-9-THC was significantly related to speeding but not to driving errors. In relation to driving errors, those with delta-9-THC levels of 5.00 ng/mL or more were less likely (41% less) to make errors than those who tested negative. These authors conclude that cannabis predicts risky driving behaviors in fatal accidents, although alcohol has a greater weight in the prediction of these behaviors.

Keric et al. [45] approached the study of marijuana legalization and alcohol consumption through the presence of traffic accident-related injuries. For this purpose, they interviewed surgeons from trauma units in hospitals in Texas and California between the years 2006 and 2012. The results of the study indicate that in those states where marijuana is legal, 90% of the surgeons do not find an increase in injuries associated with traffic accidents. With regard to the presence of cases with injuries resulting from a traffic accident, 4% tested positive for marijuana, 21% for alcohol, and 3% for both substances in the state of Texas. In California, 23% of the cases tested positive for marijuana, 50% for alcohol, and 7% for both substances. After the legalization of marijuana in 2010, there was no increase in the number of cases of trauma related to traffic accidents compared to before legalization.

Along the same lines, Kruse et al. [44] aimed to determine the impact of marijuana legalization on drug and alcohol detection in traffic accident victims in six American states between 2006 and 2018. In all states, there was an increase in the incidence of detecting THC in blood. These increases were 9.5% in Arizona and 5.4% in California, the state with the highest incidence percentages, going from 20.8% to 26.2%. However, this change was not significant, with 5.9% in Ohio, 3% in Oregon, 2.3% in New Jersey, and 15.3% in Texas, a state where marijuana is not legalized, this being the state with the greatest change in incidence, from 3% to 18.3%. With respect to alcohol, there was no change over time in most states. The study points out that there is no relationship between marijuana legalization and the probability of detecting THC in urine after a hospital admission related to a traffic accident, nor with alcohol consumption.

Pollini et al. [47] focused on the effect of marijuana decriminalization on the detection of THC-positive drivers and the number of fatal traffic accidents in the state of California. The results of the study indicate that after decriminalization, there is a significant increase in the number of drivers involved in fatal crashes who tested positive for THC, although this increase does not occur in drivers who drive on weekend nights. Finally, Hamzeie et al. [46] focused mainly on the incidence of drivers testing positive for cannabis after being involved in a traffic accident, differentiating between states that have legalized the use of marijuana, those with laws on decriminalization of its use, and those that have not legalized either use or possession. The results of the study show that in states where there are laws on legalization or decriminalization of cannabis use and possession, there is a higher probability that the driver will test positive for THC (48%), compared to those states that have not legalized its use or possession (17%). In addition, they point out that drivers who have tested positive for cannabis have more serious injuries and are younger; specifically, for each year of age, the probability of testing positive for cannabis decreases by 3%. In the case of women, the probability is reduced by 22%, and those with a valid driving license, by 14%. The study also shows a positive relationship between cannabis and alcohol consumption, such that for every 1 g/dL of alcohol, the probability of testing positive for cannabis increases by 150%.

## 4. Discussion

The objective of the present systematic review was to assess the impact of cannabis legalization on traffic fatalities. When looking at the legalization of medical cannabis, two papers found increases in the number of fatal accidents after legalization [22,24]. Other papers report increases in some of the states included in the study [26,27,28], while other studies point to the role of medical cannabis dispensaries in the rise in traffic accidents after legalization [25,28]. For their part, two studies [29,43] found no increase in traffic accidents after legalization. These results partially accept the first of the study’s hypotheses, which pointed to a positive relationship between the legalization of medical cannabis and the prevalence of traffic accidents.

Something similar happened with respect to the study’s second hypothesis, which pointed to a relationship between the legalization of recreational cannabis and traffic accidents. Three studies [35,37,39] found an increase in traffic accidents, while nine papers report an increase in traffic accidents after legalization or between states that have legalized recreational cannabis and those that have not [3,31,32,33,34,36,38,40,41].

For their part, Pollini et al. [47] found an increase in cases where cannabis use is present in fatal traffic accidents following legalization. However, the study does not specify whether this is with respect to medical or recreational cannabis.

With respect to visits to hospitals for injuries after a traffic accident, three studies did not find an increase in the number of cases [30,44,45], while one study [36] did find an increase in the number of cases [30,44,45] of traffic accident-related injuries following legalization.

To summarize, as shown in Table 7, there is a greater number of articles that have found scientific evidence supporting the impact of cannabis legalization or decriminalization on road safety, both in the comparison between states or countries and in the pre-legalization, decriminalization, and post-legalization periods of medical and/or recreational cannabis.

The study’s third hypothesis proposed a positive relationship between cannabis legalization and a greater number of risk behaviors associated with driving. The results of the studies lead to the acceptance of this hypothesis. Couper and Peterson [48], Rotermann et al. [35], Lensch et al. [33], and Benedetti et al. [42] report an increase in the number of cases of drivers who drive after consuming. Lensch et al. [33] found a higher prevalence after the legalization of riding as a passenger of a driver who had consumed cannabis, while Rotermann et al. [35] found a lower prevalence of this behavior. Regarding protective driving behaviors such as wearing a seat belt or wearing a helmet on a motorcycle, Lensch et al. [33] point to a higher use after legalization, while Borst et al. [36] and Steinemann et al. [24] indicate a lower use of these measures. Woo et al. [8] point to a relationship between testing positive for THC and fatal driving errors as well as increased speeding, behaviors that lead to a higher probability of a fatal traffic accident, while Hamzeie et al. [46] point to a higher probability of an accident after legalization.

The fourth hypothesis, about identifying risk factors among drivers who test positive for cannabis, is accepted. Among the risk factors most commonly identified are being male [8,35,46,48], being a young person between 18 and 30 years old [8,24,35,36,46,48], and the use of alcohol in conjunction with cannabis [3,8,22,34,46]. Another risk factor is driving a motorcycle [8,36], although, as noted by Nazif-Muñoz et al. [34] for Uruguay, motorcyclists have lower death rates in traffic accidents where cannabis is involved than drivers of other types of vehicles, while Steinemann et al. [24] also noted driving at night as a risk factor.

The presence of marijuana dispensaries may be another risk factor in the increase of traffic accidents if the work of Sevigny [25] with respect to MCL and that of Aydelotte et al. [40] with respect to RCL are taken into account.

As protective factors for preventing traffic accidents related to marijuana use, Callaghan et al. [30] point to an increase in penalties for driving under the influence of this substance, and Benedetti et al. [42] point to the need to regulate the permitted consumption rates for drivers.

When interpreting the results of the studies analyzed, the limitations of these studies, many of them significant, should be taken into account, such as the number of states participating [3,8,30,31,39,40,46,47], the type of studies that do not allow causality to be established [26,32,33,35,43], and mainly laws about decriminalization and/or legalization of marijuana between states, and differences among states regarding THC detection limits [25,36,37,42,43,44,46], or the collection of data through self-reports [23,35,42], among others.

Regarding the limitations of the present study, we only selected articles written in English or Spanish, and we did not include non-English research carried out in countries such as Canada, where French is also spoken. Another limitation refers to the databases used, which were Web of Science and Scopus only. On the other hand, future work should address the longitudinal impact of marijuana legalization in those states that have legalized it, controlling for a greater number of variables. In addition, it may be important to find out what role marijuana dispensaries play with respect to traffic accidents, as well as the establishment of laws on the levels of consumption allowed when driving.

Among the practical implications, it is worth mentioning, first, the reflection to be made by those countries or states that have not legalized marijuana and are considering its legalization, taking into account not only economic factors but also aspects directly or indirectly related to health. On the other hand, the presence of risk factors in traffic accidents involving cannabis makes it possible to develop communication campaigns and prevention programs aimed at controlling these variables or risk groups, focusing on not consuming if driving, not accompanying a driver who has consumed, or the use of safety mechanisms when driving [49].

## 5. Conclusions

The results of the studies are not conclusive, although a greater number of studies (22 articles) [3,8,22,23,24,25,26,27,28,30,31,32,33,34,36,38,40,41,43,46,47,48] show a negative effect of the legalization or decriminalization of cannabis on road safety, mainly on the increase in traffic accidents after the legalization or in some of the states that participated in the studies. By contrast, only seven studies [29,30,35,37,39,44,45] show no increase in traffic accidents or in the number of visits to hospitals following an accident. Regarding attitudes and risk behaviors associated with driving after consumption, it can be concluded that these behaviors are more prevalent if we consider that nine studies report some type of risk behavior compared to only one study that reports a greater use of protective measures after legalization. Finally, we can make inferences about different risk factors in traffic accidents associated with cannabis consumption, such as being male, young, and having also consumed alcohol.

## Figures and Tables

**Figure 1 ijerph-20-04655-f001:**
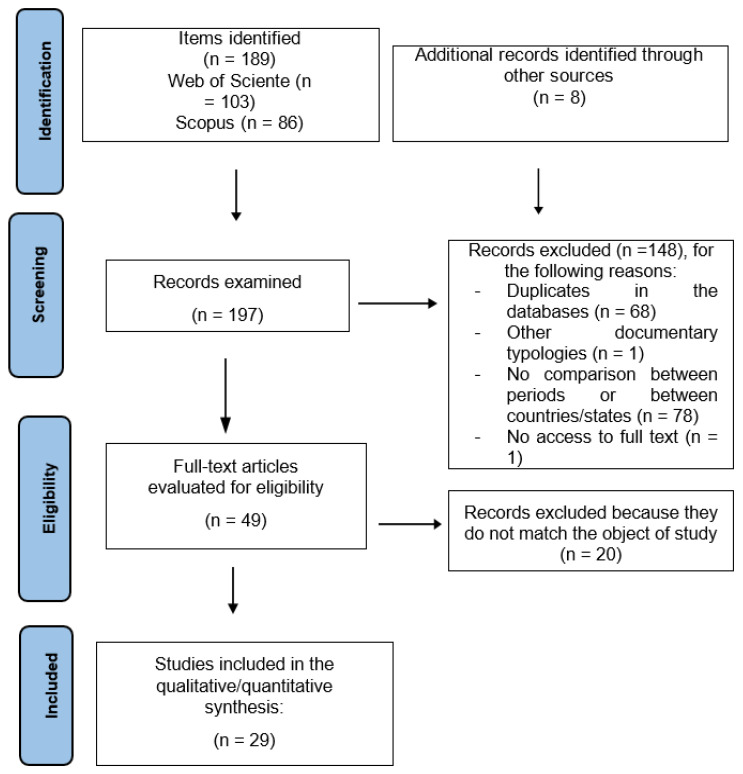
Flow of information through the different phases of a systematic review.

**Table 1 ijerph-20-04655-t001:** Studies related to the legalization of medical cannabis: country, sample, and most relevant variables.

Article	Country (States)/Years	Sample Size	Study Group	Variables	Data Collected/Detection Method
Jones et al. (2019) [22]	U.S. (Arizona) (2008–2014)	THC+: N = 2590 Alcohol+: N = 5266 THC+ and alcohol: N = 1086	Injured driver	Use of cannabis, alcohol, or both substances after a traffic accident. Pre-legalization: January 2008–April 2011; Post-legalization: April 2011–December 2014.	Urine drug screens for cannabis metabolites and BAC.
Wadsworth and Hammond (2019) [23]	Canada, England, and U.S. July, 2017	Canada: N = 4008 England: N = 3970 U.S.: N = 4086	Youth	Use, access, perceptions of harm and driving after cannabis and/or alcohol use. Sociodemographic variables.	Self-report completed using web-based surveys.
Steinemann et al. (2018) [24]	U.S. (Hawaii) (1993–2015)	Pre-legalization (1993–2000): N = 560; THC+ 31 (6%). Post-legalization (2001–2015): N = 1018. THC+ 151(15%).	Drivers killed	THC+ and alcohol in drivers. Pre- and post-legalization.	FARS. Urine toxicology and blood drug test.
Sevigny (2018) [25]	U.S. (1993–2014)	Nearly 1.2 million drivers involved in fatal traffic accidents. THC+ 8.8% BAC ≥ 0.08 = 20.1%.	Drivers	Presence of THC, other illicit drugs, and alcohol. Variables related to cannabis regularization. Contextual and control variables.	FARS. Blood drug test.
Santaella-Tenorio et al. (2017) [26]	U.S. (50 states) (1985–2014)	N = 1,220,610.	Deaths in traffic accidents	Fatal traffic accidents. Age 14–24, 25–44, and 45< years. Date MML by State. Dispensaries.	FARS. Blood drug test.
Masten and Guenzburger (2014) [27]	U.S. (14 states that have legalized cannabis and 37 jurisdictions without MCL) (1992–2009)	Drivers involved in fatal crashes: THC+ N = 19,977.	Drivers	Presence of THC in drivers involved in fatal traffic accidents.	FARS. Blood and/or urine drug test.
Salomonsen-Sautel et al. (2014) [28]	U.S. (Colorado and 34 states without MCL) (1994–2011)	Proportions of drivers involved in fatal crashes. Colorado: THC+ between 4.5% to 10%. States without MCL: THC+ between 1.1% to 4.1%	Drivers	THC+ or alcohol BAC ≥ 0.08. Pre-commercialization, June 1994–July 2009; Post-commercialization, July 2009–2011.	FARS. Blood and/or urine drug test.
Anderson et al. (2013) [29]	U.S. (1990–2010)	Age: 15–60< years old. 20.48 (7.15)	Fatalities per 100,000 people	Fatal traffic accidents. Sociodemographic variables. Control states and MCL states.	FARS. Blood drug test.

Note: BAC (blood alcohol concentration), FARS (Fatality Analysis Reporting System database), and MCL (medical cannabis legalization).

**Table 2 ijerph-20-04655-t002:** Studies related to the legalization of medical cannabis: design, main results, and limitations.

Article	Statistical Analysis	Main Result	Limitations
Jones et al. (2019) [22]	Regression analysis of interrupted time series.	Before legalization, decreased positive drivers by 0.5%/year (95% CI: −1.0/year, 0.0/year). After legalization, a significant increase in the trend of cannabis-positive drivers of 0.6%/year (95% CI: 0.0/year, 0.8/year).	Differences between and within states in drug testing protocols. Testing positive for THC does not imply recent use. No control group.
Wadsworth and Hammond (2019) [23]	Chi-squared tests, nominal logistic regression, and multinomial logistic regression models.	There are differences in driving after cannabis use between countries. England: less likely to drive after drinking than Canada (*p* < 0.001) and the U.S. (*p* < 0.001). Canada: less likely than the U.S. (*p* < 0.001).	Self-report measures. Social desirability. Biases in a recall. General legislation of the country is taken and differences between different districts/cities are not taken into account. Non-random subject selection.
Steinemann et al. (2018) [24]	*t*-tests and Chi-squared tests.	Significant increase (*p* < 0.001) in the number of drivers testing positive for THC after legalization in 2000 in fatal accidents.	THC+ does not imply recent consumption. A THC level cannot be extrapolated to an accident risk level. There is no discrimination between acute, heavy, and chronic use.
Sevigny (2018) [25]	Generalized linear model (GLM) with a binomial distribution and logit link function. Sensitivity analyses.	There is no relationship between the MCL and the number of fatal traffic accidents OR [95% CI] 1.05 [0.93, 1.19]. The implementation of medical cannabis dispensaries does correlate with a higher number of accidents involving cannabis OR [95% CI] 1.14 [1.02, 1.29] (*p* < 0.01).	THC+ does not imply recent use. Differences between states in drug testing protocols, as well as changes in trends after legalization. Not all crash victims are drug tested. Levels of cannabis use are not systematically quantified.
Santaella-Tenorio et al. (2017) [26]	Multilevel regression models with state-level random intercepts.	States with MCL laws have lower rates of traffic fatalities than states without MCL laws (26.3% lower; 95% confidence interval [CI] = 13.9%, 36.9%). A 10.8% reduction in fatalities after MCL (95% CI = 9.0%, 12.5%; % reduction = [1 − exp(−0.114)] − 100). Following the legalization of cannabis, there was a reduction in fatal traffic accidents in 7 states.	Causal relationships cannot be established. Local aspects that may have an influence are not taken into account. Short post-legalization periods. THC tests are not performed.
Masten and Guenzburger (2014) [27]	Time series analyses. Auto-regressive integrated moving average analysis.	MCL is only associated with an increase in fatal traffic accidents in California, Hawaii, and Washington (out of a total of 12 states) after adjusting for the frequency of drug testing by state and the prevalence of cannabinoids in drivers in states without legalization laws.	They only study fatal traffic accidents. State differences in drug testing protocols and changes in trends after legalization. Not all crash victims are tested for drugs. THC+ does not imply recent use.
Salomonsen-Sautel et al. (2014) [28]	Linear regression analysis. Estimated generalized least squares (EGLS) methods.	Significant trend change (2.16 (0.45), *p* < 0.0001) in Colorado (not in the rest of the states) after legalization, with an increase in THC-positives among drivers involved in fatal accidents.	They only study fatal traffic accidents. THC+ does not imply recent use. Levels of marijuana use are not systematically quantified.
Anderson et al. (2013) [29]	Linear regression analysis by ordinary least squares (OLS) estimates	Legalization leads to a significant decrease in fatal traffic accidents, although when state-specific time trends are included, the decrease is not significant (*p* = 0.139). Legalization leads to a reduction in crashes where the driver tests positive for alcohol (R = −0.141, *p* < 0.01) and in cases where the alcohol level is above 0.10 (R = 0.168, *p* < 0.05).	Not specified

**Table 3 ijerph-20-04655-t003:** Studies related to the legalization of recreational cannabis: country, sample, and most relevant variables.

Article	Country (States)/Years	Sample Size	Study Group	Variables	Data Collected/Detection Method
Callaghan et al. (2021) [30]	Canada (Alberta and Ontario) (2015–2019).	Alberta: N = 52,752. Youth-driver N = 3265, Ontario: N = 186,921. Youth-driver N = 4565.	Injured drivers	Weekly visits to medical emergency units for injuries in traffic accidents. Young drivers and total drivers. Pre-legalization: 1 April 2015–16 October 2018. Post-legalization: 17 October 2018–31 December 2019.	National Ambulatory Care Reporting System database.
Tefft and Arnold (2021) [31]	U.S. (Washington). (2008–2019).	THC+: N = 735; negative: N = 3528; unknown: N = 4019	Drivers involved in fatal crashes	THC in blood: positive (1.0 ng/mL or more). Negative no THC detected in blood. Unknown: no drug test or no confirmation. THC concentrations were 0, 1.0–4.9 ng/mL, 5.0–9.9 ng/mL, and ≥10.0 ng/mL. Pre-legalization: 1 January 2008–5 December 2012. Post-legalization: 6 December 2012–31 December 2019. Vehicle and crash characteristics.	FARS. Blood drug test.
Windle et al. (2021) [32]	U.S. (10 states and District of Columbia). RCD (7 states). (2007–2018).	Periods RCL: 17,116 accidents and 18,580 deaths. Periods without legalization: 56,866 accidents and 61,822 deaths.	Death from motor vehicle collision	Fatal accidents. Deaths following a collision. Recreational cannabis dispensaries (7 states). Cannabis-specific impaired driving law: zero tolerance, per se limit ≥ 2 ng/mL THC, per se limit ≥ 5 ng/mL THC, or none.	FARS.
Lensch et al. (2020) [33]	U.S. (2018)	N = 17,112 adults. Legal sale states: N = 5548. Non-legal sale states: N = 11,564.	Drivers and Passengers	Six legal sale states with dispensaries. Non-legal sale states where despite legalizing marijuana, there are no dispensaries. Attitudes and behaviors related to driving, consumption, and frequency of cannabis use.	Web-based surveys (Self-reported).
Nazif-Muñoz et al. (2020) [34]	Uruguay (Montevideo and 4 rural provinces) (2012–2017).	N = 3037 fatal accidents.	Drivers and motorcyclists	Accidents and mortality rate in traffic accidents. Urban and rural areas. Type of vehicle: automobiles and motorcycles.	National Road Safety Agency. Ministry of Transport and Public Works
Rotermann (2020) [35]	Canada. (2018–2019).	Pre-legalization: N = 17,683. Post-legalization: N = 21,872.	Drivers and passengers	Pre-legalization: first, second, and third quarters of 2018. Post-legalization: first, second, third, and fourth quarters of 2019. Consumption and origin of the product. Driving a vehicle 2 h after consumption. Accompanying a driver who has consumed.	National Cannabis Survey: Internet-based electronic questionnaire
Borst et al. (2020) [36]	U.S. (California). (2010–2018).	THC+: N = 1345. THC+ and Alcohol+: N = 578. No use: N = 7078.	Patients	Type of use: no use, cannabis only, alcohol only (>0.08%), cannabis and alcohol, and cannabis with methamphetamines or cocaine. Pre-legalization: 2010–2015. Post-legalization: 2016–2018. Data concerning injuries and protective measures.	San Diego County’s trauma center registries. Blood and urine test.
Hansen et al. (2020) [37]	U.S. (Colorado, Washington, and control states) (2000–2016).	Not specified	Drivers	Fatal traffic accidents, alcohol-involved accidents, and marijuana-involved accidents.	FARS.
Santaella et al. (2020) [38]	U.S. (Colorado, Washington, and 42 control states) (2005–2017).	Not specified	Driver and passenger deaths	Mortality rate per traffic accident and year. Vehicle miles traveled. Age adjusted fatality rates. Pre-legalization: 2007–2013. Post-legalization: 2014–2017.	FARS.
Delling et al. (2019) [3]	U.S. (Colorado and control states). (2010–2014).	Colorado: N = 2,088,909. New York: N = 11,726,283. Oklahoma: N = 2,334,988.	Patients	Admission to hospital due to traffic accidents. Sociodemographic and health variables and alcohol consumption.	Healthcare Cost and Utilization Project database.
Aydelotte et al. (2017) [39]	U.S. (Colorado, Washington, and 8 control states) (2009–2015).	N = 60,737 fatal traffic accidents.	Drivers and passengers	Number of fatal traffic accidents. Billion vehicle miles traveled.	FARS.
Aydelotte et al. (2019) [40]	U.S. (Colorado, Washington, and 9 control States) (2007–2017).	Fatal accidents (Pre-legalization N = 12,348; Post-legalization N = 12,865).	Driver and passenger deaths	Fatal traffic accidents. Legalization: Before November 2007–October 2012, and January 2013–December 2017. Period of commercial dipensaries available: August 2014–December 2017. Period of the opening of the first dispensary: August 2010–December 2013. Billion vehicle miles traveled. Gross domestic product. Control variables in control states.	FARS.
Lane and Hall (2019) [41]	U.S. (12 states) (2009–2016)	States LRC (Colorado, Washington, and Oregon) and 9 adjoining states.	Deaths in traffic accidents	Rate of fatal traffic accidents per million population. States that have legalized recreative cannabis and neighboring states.	Centers for Disease Control and Prevention’s Wide-Ranging Online Data for Epidemiologic Research and RoadSafetyBC report.

Note: FARS (Fatality Analysis Reporting System database), MCL (medical cannabis legalization), and RCD (recreational cannabis dispensaries).

**Table 4 ijerph-20-04655-t004:** Studies related to the legalization of recreational cannabis: design, main results, and limitations.

Article	Statistical Analysis	Main Results	Limitations
Callaghan et al. (2021) [30]	Interrupted time-series analysis. Seasonal autoregressive integrated moving average models	There is no significant effect of increased injury visits among young drivers and all drivers before and after legalization. Significant differences (95% CI −26.32; 84.19; *p* = 0.30) are observed for all drivers in Ontario in emergency department visits.	Data are only collected from two provinces in Canada and from cases that have resulted in moderate to severe injuries.
Tefft and Arnold (2021) [31]	Logistic regression and marginal standardization.	Increase in the proportion of drivers who tested positive for THC from 9.3% before and 19.1% after legalization (APR: 2.3, 95% CI: 1.3, 4.1) and in the concentration of THC (APR: 4.7, 95% CI: 1.5, 15.1).	There is a significant number of drivers for which there is no drug test. Bias in the results by not taking into account other variables. Data are not compared with other states that have not legalized RC.
Windle et al. (2021) [32]	Poisson regression, meta-analyzed estimates, and DerSimonian and Laird random-effects models.	Increase in fatal traffic accidents (IRR 1.15, 95% CI 1.06–1.26) and deaths (IRR 1.16, 95% CI 1.06–1.27) in the first year after legalization.	Observational study. Jurisdiction differed among states.
Lensch et al. (2020) [33]	Chi-square tests. APR and 95% CI.	Higher incidence of use in states that legalized cannabis in the previous 30 days (APR: 1.34; 95% CI: 1.19, 1.51) and in the previous 12 months (APR: 1.16; 95% CI: 1.06, 1.28). Higher protective behaviors in states that have legalized RC.	Cross-sectional study. Non-representative sample of the general population.
Nazif-Muñoz et al. (2020) [34]	Interrupted time-series analysis. Extension autoregressive integrated moving average.	Significant increase in the light motor vehicle driver fatality rate (CI = 11.6, 93.3, *p* = 0.012). Significant increase in automobile driver mortality in Montevideo (CI = 0.01, 0.11, *p* = 0.025) but not in rural areas.	Prevalence of cannabis use in traffic accidents. Accidents with injuries are not considered. Possible biases in the coding of accidents.
Rotermann (2020) [35]	*t*-test statistics.	Stability in the number of cases of driving after having consumed before and after legalization, being more frequent in men than in women (*p* < 0.05). In general, decrease in the number of cases of traveling in a vehicle whose driver had consumed.	Self-report data. The type of design does not allow for causal inferences. The study is limited to surveyed households only.
Borst et al. (2020) [36]	Multivariate logistic regression. Linear regression. Binomial logistic regressions. Pearson χ^2^. Time-series regression analysis	A 7.6-percentage point increase of THC+ cases in accidents after legalization. The THC+ group used fewer protective measures while driving (8.5% vs. 14.3%, *p* < 0.001) and suffered more serious injuries (8.4 ± 9.4 vs. 9.0 ± 9.9, *p* < 0.001) than the non-consumption group.	There is no legal threshold for driving under the influence of cannabis. Time in which cannabinoids are maintained in the blood. No toxicological screening of all patients. Variability in detection rates between institutions.
Hansen et al. (2020) [37]	Synthetic control approach. Permutation testing of the ratio of mean squared error ratios	Control states show the same increases in the number of accidents per billion vehicle miles traveled in Colorado (*p* = 0.361) and Washington (*p* = 0.404).	Levels of marijuana use not quantified and variation among states. Only fatal traffic accidents. No causality can be asserted.
Santaella et al. (2020) [38]	Ecological study used a synthetic control approach. Mean square prediction error.	Increase (*p* = 0.047) in traffic fatalities in Colorado but not in Washington following legalization of RC.	Other variables associated with legalization. Time of legalization in Washington. No data on injuries. Not included if the driver tested positive for THC at the time of the accident.
Delling et al. (2019) [3]	Linear, logistical, and multinomial models.	An increase (*p* < 0.05) in the number of traffic accidents, alcohol abuse, overdose injuries and a decrease in chronic pain admissions is observed following the legalization of RC versus control states.	The post-legalization period is short (two years). They only compare with two control states. Colorado legalized medical marijuana prior to 2012.
Aydelotte et al. (2017) [39]	Random effects multivariate regression for panel data. Difference-in-differences approach.	Despite an increase, there is no significant association between legalization and changes in the rate of fatal traffic accidents (ADDC = +0.2 fatalities/billion vehicle miles traveled; 95% CI = –0.4, +0.9) in the first 3 years of legalization.	They only analyze two states in which the substance has been legalized. They do not study adjacent control states. They study fatal traffic crashes as a whole, not those in which cannabis is involved.
Aydelotte et al. (2019) [40]	Retrospective longitudinal cohort study. Difference–indifference analyses.	Significant increase in the number of fatal traffic accidents in states that legalized RC after opening dispensaries to buy recreational marijuana (CI: +0.4 to +3.7, *p* = 0.020) compared to control states. This increase was not significant before the dispensaries opened (CI: −0.6 to 2.1, *p* = 0.087).	Do not make any statistical adjustments for multiple analyses. Biases in the selection of control states. Missing more states where recreational marijuana use has been legalized. Only study fatal traffic accidents.
Lane and Hall (2019) [41]	Interrupted time-series design. Generalized least squares regression models.	The general trend is an increase in the mortality rate both in states with legalization laws and in neighboring states. There is a trend of increased mortality (*p* < 0.001) at the beginning and a decrease in the subsequent months (*p* < 0.001). At 6 months these effects are greater (step: 1.36, *p* = 0.006; trend: −0.07, *p* < 0.001).	They do not differentiate between fatal traffic accidents in which marijuana is involved and those in which it is not. They do not consider other types of accidents. They do not analyze economic factors or current policies of neighboring states regarding cannabis.

**Table 5 ijerph-20-04655-t005:** Studies related to the legalization of recreational and medicinal cannabis and its decriminalization: country, sample, and most relevant variables.

Article	Country (State)/Years	Sample Size	Study Group	Variables	Data Collected/Detection Method
Benedetti et al. (2021) [42]	U.S. (2013–2017).	CR illegal: N = 10,294 drivers; CR legal: N = 876 drivers. MC illegal: N = 5782 drivers; MC legal: N = 5388 drivers.	Drivers	Marijuana use while driving in the last year. States with the legalization of MM, MM, and MR or no legalization of both. Marijuana policy. Sociodemographic variables. TSCI is a nationally representative annual survey.	Self-reported.
Kruse et al. (2021) [44]	U.S. (Arizona, California, Ohio, Oregon, New Jersey, and Texas) Pre-legalization: 2006–2012. Post-legalization: 2013–2018.	Not specified	Patients with trauma	Vehicle collisions. THC-positive patients and alcohol-positive patients > 0.08 g/dL).	Data from different universities. Urine and blood analysis.
Woo et al. (2019) [8]	U.S. (Washington) (2008–2016).	N = 10,155 accidents and 5931 drivers. THC > 5.00 = 4.2%, THC <5 = 3.1%, Clean 92.6%. BAC > 0.08 = 19.2%, BAC <0.08 = 3.5%, Clean 77.3%.	Drivers	Fatal accidents. Speeding and driver error. THC (<5 ng/mL in blood or >5 ng/mL). Carboxy-THC. Alcohol (0.08 less or more). Control variables. Environmental Contexts.	FARS. Blood tested.
Keric et al. (2018) [45]	U.S. (Texas and California) (2006–2012).	N = 127 Surgeons. UTHSCSA center: patients traffic accidents N = 7171. Alcohol+ 21%, THC+ 4%, alcohol and THC+ 3%. Center in California: N = 16,084, alcohol+ 50%, THC+ 23%, alcohol and THC+ 7%.	Surgeons and patients	Alcohol (>0.08 g/dL). Marijuana (>50 ng/mL in Texas and >100 ng/mL in California). Injuries related to traffic accidents. Decriminalization in CA in 2010. Electronic survey completed by the members of the American Association for the Surgery of Trauma.	Trauma center registries at The University of California Irvine and UTHSCSA.
Lee et al. (2018) [43]	U.S. (16 states) (2008–2015).	Number of accidents involving cannabis in states with law changes. Before, N = 1458. After, N = 938.	Drivers	Types of states according to cannabis legalization and decriminalization laws.	FARS.
Hamzeie et al. (2017) [46]	U.S. (50 states and District of Columbia) (2010–2014).	THC+: N = 9301 drivers THC−: N = 65,332 drivers	Drivers	States with legalization and decriminalization laws and states with no such laws. Driver, accident, and vehicle characteristics. THC+ and THC−.	FARS.
Pollini et al. (2015) [47]	U.S. (California). (2008–2012).	(2008–2010): N = 1718; THC+ N = 203 (2011–2012): N = 1142; THC+ N = 175	Drivers	Detection of cannabis use in drivers involved in traffic accidents. Decriminalization period: 2011–2012. No decriminalization period: 2008–2010.	FARS.
Couper and Peterson (2014) [48]	U.S. (Washington, D.C.) (2009–2013)	N = 25,179, age 14–85 years old. Median age 25 years pre-legalization and 26 years post-legalization.	Drivers	Detection of THC consumption. (Pre-legalization THC+ 0.2 ng/mL, carboxy-THC of 0.10 ng/mL). Pre-legalization: 2009–2012. Post-legalization: 2013.	Blood tested

Note: FARS (Fatality Analysis Reporting System database).

**Table 6 ijerph-20-04655-t006:** Studies related to the legalization of recreational and medicinal cannabis and its decriminalization: design, main results, and limitations.

Article	Statistical Analysis	Main Results	Limitations
Benedetti et al. (2021) [42]	Multiple logistic regression model.	States with MCL have a higher number of drivers who have driven under the influence of marijuana versus states that have not legalized MC and/or RC (OR 1.29; 95% CI 0.98, 1.70; *p* = 0.075). THC threshold laws: less likely to drive after consumption (OR 0.74; 95% CI 0.57, 0.95; *p* = 0.018).	Biases associated with self-reports. Quasi-experimental design that does not allow inferring causal relationships between marijuana use among drivers and states’ policies on marijuana use.
Kruse et al. (2021) [44]	Retrospective analysis of data. Percentages.	There seems to be no relationship between legalization and the probability of finding THC in patients admitted after an accident.	Discrepancies in urine THC detection limits by institution. A lack of standardized laws by the state does not allow the detection of real THC prevalence.
Woo et al. (2019) [8]	Series of logistic regressions.	Being a young man, driving a motorcycle, and testing positive for alcohol, delta 9-THC, or carboxy-THC and other drugs (*p* < 0.001) are risk factors for speeding. Cannabis predicts risky driving behavior.	Only fatal accidents are examined. Washington State data only. Not all crashes tested for drugs. Measurement errors in drug rates.
Keric et al. (2018) [45]	Time frame. Percentages.	A total of 90% of surgeons report no increase in cases of traumatic injuries in traffic accidents after cannabis legalization.	Not specified.
Lee et al. (2018) [43]	Series estimation of crash modification factors.	Increase in fatal accidents in which the driver tests positive for cannabis, mainly with decriminalization (*p* < 0.001) and/or legalization of RC but not MC (*p* < 0.001) Other effects are between decriminalization and decriminalization and MCL (*p* = 0.020) and between MCL and full legalization (*p* = 0.010).	Short post-legalization periods. Differences between states in drug testing protocols and trends after legalization. Difficulties in selecting a control group. Cannot assert causality.
Hamzeie et al. (2017) [46]	Logistic regression models	Higher probability of testing positive for THC in an accident in states with cannabis decriminalization (17%) and/or legalization laws (48%) (*p* < 0.001). Being young, male, positive for alcohol, and exhibiting more risky driving behaviors increased the probability of THC+ (*p* < 0.001).	They only test for CRL in two states and for a short period of time. Not all drivers take the drug test. Differences between states in drug testing protocols and trends after legalization.
Pollini et al. (2015) [47]	Multiple logistic regression analyses	Significant increase in the prevalence of cannabis positives among drivers involved in fatal crashes after decriminalization (17.8%; 95% CI: 14.6, 20.9). No change in THC positives among weekend nighttime drivers after decriminalization (9.2%; 95% CI: 6.3, 12.2).	Differences in drug testing protocols. Changes in consumption trends after legalization. Small and restricted sample. THC+ does not imply recent use.
Couper and Peterson (2014) [48]	Chi-squared tests	After a stable trend, there is a significant increase in the percentage of positive cases of THC consumption in drivers after legalization (*p* < 0.05).	THC concentration can be altered causing problems in the cut-off point for considering a subject positive. Delays in blood collection can influence the concentration of metabolites.

**Table 7 ijerph-20-04655-t007:** List of articles used in the review according to legalization period, state or country, and effect of legalization.

Effect of Legalization/Decriminalization on Traffic Accidents	States/Countries	Periods Pre- and Post-Legalization
YES	[3,23,25,26,27,28,33,40,41,42,43,46]	[3,8,22,24,28,31,32,34,36,38,40,41,43,47,48]
NO	[29,37,39]	[30,35,39,44,45]

## Data Availability

Not applicable.

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
