# Peer review of "Effects of Cannabis Legalization on Road Safety: A Literature Review"

_ijerph, 2023, doi:10.3390/ijerph20054655_

Round 1
Reviewer 1 Report
The manuscript presents the findings from a systematic review that focused on the role of cannabis legalization (medical and/or recreational) on traffic safety. The study topic is important and the methodology appears to be strong; however the results text as currently presented is a major weakness. Specifically, the authors provide no synthesis or summary information overall for the findings, nor for individual sections. Instead, the findings of each study are simply recounted, one per paragraph, across all sections. Thus, from reading the results as presented, it is largely unclear to this reviewer what the findings were and how they align with each other (or don't) (in this regard, the presentation of the results requires the reader to consolidate and synthesize the information for themselves, which is burdensome and not best practice for scientific writing). Because the results were presented piece by piece, the discussion was also somewhat hard to follow, as most of the discussion of the findings was first summarized there. I recommend that the authors make a substantial effort to reorganize the results section to clearly describe the overall findings, and then present a summary of the findings for each of the three topical sections, to synthesize the results for the reader. A separate paragraph should summarize the findings from the review of methodological quality, and describe the overall quality of the literature reviewed. Then in the discussion, the comparison with existing literature, etc should be more well integrated. Besides this major concern, another more minor observation is that the authors use the terms cannabis and marijuana interchangeably throughout the manuscript; I recommend using one single term (preferably the term cannabis) throughout.
Author Response
REVIEW: The manuscript presents the findings from a systematic review that focused on the role of cannabis legalization (medical and/or recreational) on traffic safety. The study topic is important and the methodology appears to be strong; however the results text as currently presented is a major weakness. Specifically, the authors provide no synthesis or summary information overall for the findings, nor for individual sections. Instead, the findings of each study are simply recounted, one per paragraph, across all sections. Thus, from reading the results as presented, it is largely unclear to this reviewer what the findings were and how they align with each other (or don't) (in this regard, the presentation of the results requires the reader to consolidate and synthesize the information for themselves, which is burdensome and not best practice for scientific writing). Because the results were presented piece by piece, the discussion was also somewhat hard to follow, as most of the discussion of the findings was first summarized there. I recommend that the authors make a substantial effort to reorganize the results section to clearly describe the overall findings, and then present a summary of the findings for each of the three topical sections, to synthesize the results for the reader. A separate paragraph should summarize the findings from the review of methodological quality, and describe the overall quality of the literature reviewed. Then in the discussion, the comparison with existing literature, etc should be more well integrated. Besides this major concern, another more minor observation is that the authors use the terms cannabis and marijuana interchangeably throughout the manuscript; I recommend using one single term (preferably the term cannabis) throughout.
ANSWER: As the reviewer suggests, before describing the results of each of the studies in more detail, a more general synthesis is made in the first paragraph in relation to the number of studies that find a relationship between legalization and those that do not. find such a relationship.
In response to the reviewer's comment, the results section has been reorganized, including a first subsection of general aspects of the studies, including information on the samples or on the main source of data collection.
The discussion section has been reorganized and the term cannabis has been replaced by marijuana in a large part of the manuscript, although in some cases it has not been replaced because the authors of the original manuscript used that term.
Reviewer 2 Report
General comments
The subject of the paper is of very high interest
However, I think there is a need for important changes in the presentation
Table 1 : ML, MML, RM =? We can guess but please specify
please provide much more information on each study. Please be much more precise. We need to really understand how the study was conducted (type of study), what is the study population ? sample of drivers?, all injured drivers? Or a sample of? ; size of study population ? cases= who?, what are the (significant) results, etc
For instance, line 1, study of Benedetti : variables : marijuana use in driving in the last year: how is it measured? Is it a self-declarative survey ? (sounds like it given the “self-reports” in the limitations)
For instance, Line 2, study of Callaghan et al. , main results= “significant differences are observed for all drivers in Ontario”: on which variables? Compared to who ?
Back to general comments :
for before /after studies, please give the size of population (of states or countries) and years before and after , so we have an idea whether the sizes are comparable or not
For comparison between states or countries: size of population (or sample size) in one group and in the others
Give detail about the legalization in each study ;
Please provide quantitative results and only those that are significant (there is already so many results !)
Make smaller tables, by dividing the studies by type:
I suggest : use the same structure as in the results section : divide them by MM , MR and within this , divide them by type of studies :
before /after studies
geographical studies: some states versus other states
other types of studies
and within this, group them by countries?
so that it makes it easier to follow, compare, analyse. I insist that there is really a need for some structuration / hierarchy in table 1 and in the text of the results (and the same in both)
Medical cannabis : what are the dosages of the THC? I suppose medical cannabis has some controlled dosage, which may be different across countries or states
Dosage of THC in recreational cannabis? Are there some studies measuring the average amount? In some countries? States?
Please make shorter sentences
About the 19 studies (line 118) that were excluded: please provide more explanation
Part 3 : please provide more quantitative results together with their confidence intervals or p-value. Please provide only significant results (or if not, explain why you provide them)
The last 2 lines of the abstract (lines22 and23) : overstated ? we need significant quantitative results
Author Response
REVIEW: The subject of the paper is of very high interest. However, I think there is a need for important changes in the presentation.
Table 1: ML, MML, RM =? We can guess but please specify please provide much more information on each study. Please be much more precise. We need to really understand how the study was conducted (type of study), what is the study population? sample of drivers? all injured drivers? Or a sample of?; size of study population ? cases= who?, what are the (significant) results, etc
For instance, line 1, study of Benedetti: variables: marijuana use in driving in the last year: how is it measured? Is it a self-declarative survey? (sounds like it given the “self-reports” in the limitations). For instance, Line 2, study of Callaghan et al., main results= “significant differences are observed for all drivers in Ontario”: on which variables? Compared to who?
ANSWER: More information about each study is provided in the different tables that have been made after the reviewers' comments. For this, six tables are made, providing more data on the description of the sample, the main variables of each study and the source of data collection, the type of study and/or statistical analysis carried out and the main results according to the objective of the study. present systematic review. The meaning of the acronyms is specified in each table.
Back to general comments:
REVIEW: for before /after studies, please give the size of population (of states or countries) and years before and after, so we have an idea whether the sizes are comparable or not. For comparison between states or countries: size of population (or sample size) in one group and in the other. Give detail about the legalization in each study ; Please provide quantitative results and only those that are significant (there is already so many results).
ANSWER: In the tables, specifically in the sample and variables column, more information is provided about the years in relation to the pre-legalization and post-legalization period. Quantitative information regarding statistically significant differences is provided in the main results column. Of the works analysed, in this study we have focused on those that have to do with cannabis, providing the results mainly for the THC+ and/or THC+ groups in combination with other drugs, mainly alcohol.
REVIEW: Make smaller tables, by dividing the studies by type: I suggest: use the same structure as in the results section : divide them by MM , MR and within this , divide them by type of studies: before /after studies geographical studies: some states versus other states other types of studies and within this, group them by countries? so that it makes it easier to follow, compare, analyse. I insist that there is really a need for some structuration / hierarchy in table 1 and in the text of the results (and the same in both).
ANSWER: As suggested by the reviewer, Table 1 has been divided into six tables. The results section has been divided into different sections in order to better structure the information. This division is made based on whether legalization is related to medical cannabis, recreational cannabis, both, or decriminalization. Within each section, the studies related to periods, pre-legalization or òstlegalization, and those mainly related to the comparison between states are differentiated. In addition, the discussion section includes a table that includes the jobs by country or state, period, and effect of legalization.
REVIEW: Medical cannabis: what are the dosages of the THC? I suppose medical cannabis has some controlled dosage, which may be different across countries or states.
ANSWER: The results of the studies analyzed do not specify the doses of medical cannabis by state.
REVIEW: Dosage of THC in recreational cannabis? Are there some studies measuring the average amount? In some countries? States?
ANSWER: Throughout the document, the doses of cannabis in the blood are specified in relation to the laws that regulate its use, differentiating between a presence of 0.2 ng or 0.5 ng in relation to the THC+ group. In addition, among the limitations of the study, the differences between states in relation to the minimum amount under which driving is allowed are specified, although these data are not collected in the study since it is not one of the objectives of the work and because these laws they may have changed since the work was published. However, it does specify some result indicated by a reviewed article, such as the relationship between the presence of laws that regulate the amount allowed for driving and the number of traffic accidents in which cannabis is involved.
Please make shorter sentences
REVIEW: About the 19 studies (line 118) that were excluded: please provide more explanation.
ANSWER: More information is provided regarding the exclusion of these 19 articles..
REVIEW: Part 3: please provide more quantitative results together with their confidence intervals or p-value. Please provide only significant results (or if not, explain why you provide them).
ANSWER: Quantitative data of the results that were statistically significant have been provided.
REVIEW: The last 2 lines of the abstract (lines22 and23) : overstated ? we need significant quantitative results.
ANSWER: These two lines are a conclusion of the analysis of the results obtained in the reviewed articles. To justify this conclusion, in the conclusions section, it has been specified that 22 of the 29 works included in this review find evidence of the effect of legalization on road safety.
Round 2
Reviewer 1 Report
The authors were overall responsive to my prior suggestions and the manuscript overall is greatly improved. I have some remaining suggestions, detailed below.
1) The authors should specify the date that the literature review was completed.
2) Line 158-159 are not clear. There is a reference to the data and 16 studies using a specific source; however it is not clear what this refers to, given the total study N was 29 studies.
3) Tables 1-4: the information included in both columns should be separated into more columns, instead of presented with semicolons, which make the table difficult to read.
4) At various points the authors switch from past to present tense when describing study findings; tense should be consistently used throughout.
5) Discussion section- the first sentence is too long and difficult to follow.
6) The discussion section still seems long and largely recounts the results, instead of synthesizing and summarizing the findings and main take home points overall.
7) Line 615 says only English language articles were selected, which contrasts statements in the methods section that English and Spanish language articles were included in the review. please clarify
Author Response
Reviewer 1
Comments and Suggestions for Authors
The authors were overall responsive to my prior suggestions and the manuscript overall is greatly improved. I have some remaining suggestions, detailed below.
Comment 1) The authors should specify the date that the literature review was completed.
Answer: The day on which the search was performed has been added, specifically August 4, 2022 is included in the text.
Comment 2) Line 158-159 are not clear. There is a reference to the data and 16 studies using a specific source; however it is not clear what this refers to, given the total study N was 29 studies.
Answer: It has been clarified that of the 29 articles included in this review, 16 of them have used Fatality Analysis Reporting System (FARS) database.
Comment 3) Tables 1-4: the information included in both columns should be separated into more columns, instead of presented with semicolons, which make the table difficult to read.
Answer: A new column Study group has been added and the Sample characteristics column has been changed to Sample size, including in the latter only the number of subjects as suggested by another reviewer.
Comment 4) At various points the authors switch from past to present tense when describing study findings; tense should be consistently used throughout.
Answer: It has been revised and corrected.
Comment 5) Discussion section- the first sentence is too long and difficult to follow.
Answer: The paragraph has been reformulated.
Comment 6) The discussion section still seems long and largely recounts the results, instead of synthesizing and summarizing the findings and main take home points overall.
Answer: It has been decided to keep the discussion as it is, partly because it allows us to relate the papers to the hypotheses. In addition, it includes the limitations of the studies included in the review and practical implications.
Comment 7) Line 615 says only English language articles were selected, which contrasts statements in the methods section that English and Spanish language articles were included in the review. please clarify.
Answer: It has been corrected by including in the text …or Spanish
Reviewer 2 Report
Review of the 2Nd version of manuscript
I can see that some changes were made but some do not seem very appropriate, and as a whole they are not sufficient. It is really burdensome for the reader to read a paragraph for each study in the text: information for each study should be in the tables (so that it is easier to compare with other studies). The text itself should mostly contain a synthesis and a critical point of view.
4 papers were not included “for not being able to download through the VPN of the university of Valencia”. This is not a very scientific reason... Did you contact the authors to ask them for their paper?
Given that these are studies on aggregated data, at the level of states or countries, there should be a paragraph with the limitations of this kind of studies.
It seems that my previous request on commenting in the text only the significative results was not followed : line 180 you mention that 2 papers found an increase in so and so, but we can see in table that one of the 2 (ref 33) has confidence intervals that include zero : they are not significative…If they are not significative, this should at the very least be said in the text.
Major changes in the presentation must be made
More detailed comments :
Why did you make table1 separated from table2, since they contain the same studies? It would be more helpful to read all about each study in one big table… instead of 2
The same applies for the pair {table 3 and 4}, the pair : tables 5 and 6.
Below is a link for a DRUID deliverable :
https://www.bast.de/Druid/EN/deliverales-list/downloads/Deliverable_2_1_3.html;jsessionid=3AC071FF172D84368E679B3B49DA64F2.live11314?nn=613800
In this DRUID report, in the annex, there are a number of tables reporting different studies. They are first structured by type of objective (estimate prevalence while driving or estimate relationship between THC and risk of road crash), and then there are separated by type of studies : table 5, 6, 7. These are tables that are easy to read (even so there should have been more information in each of them). These should be used as a goal for clear tables
The content of each of your tables should much more structured.
It would help if the tables had the same structure as in a paper (introduction /material and methods / results / limitations ):
- general information first (country, years, type of study : before /after or comparison between states or else…), size of samples that are compared, but NOT including results),
- followed by a column with material (as it is now “variables and collected data),
- then by method (statistical analysis and a bit more : what is compared? Or what is estimated?),
- then by results : many more results should be here, rather than in the text (so that it is easier to compare with other studies in the table.
- column limitations : why do we find this “not specified” sometimes? limitations should also include the limitations YOU think there are.
For instance in table 1, study 33: why do you give general frequencies, and “results” in the WHOLE sample? It would be more useful to have the size of the study group before legalization, and the size of the study group after legalization. The results for each of the 2 (or more) study groups should be given in the results column (we don’t need gender and proportion of THC+ for the WHOLE sample : they are not useful for the objective of the paper). Please only provide results that are useful for the purpose of your paper
Please be 1) precise, 2) more specific, 3) more informative
For each line in the table, we need to understand, what was done (what was compared to what, and so on). It should be a good summary of the study: we should not have to go back to the text or to the article itself to understand what was done.
The distinction between before/after studies and comparison between states in not clear enough. It could help to make sub-tables so that the columns better fit the studies, and so that it is easier to compare them. This of course applies to medical, and/or recreational cannabis studies
Please provide a table summarizing the analysis of quality that you did on the selected papers. Please explain why you only use 10 indicators out of 18.
In the pdf file itself, I have added small comments on some parts. This is not at all exhaustive.
The English version needs to be checked by a native English speaker, or at least by someone who is fluent in English.
Please make shorter sentences.
The comments of the other reviewer of the first version of the manuscript do still apply

Author Response
Reviewer 2
Comment 1: I can see that some changes were made but some do not seem very appropriate, and as a whole they are not sufficient. It is really burdensome for the reader to read a paragraph for each study in the text: information for each study should be in the tables (so that it is easier to compare with other studies). The text itself should mostly contain a synthesis and a critical point of view.
Answer: We believe that including more information in the table may be difficult for readers to read. In this regard, after discussion among the authors of the paper, it was decided that it would be more appropriate to put the main aspects of the study in the table and in subsequent paragraphs comment in more detail on this articles.
Comment 2: 4 papers were not included “for not being able to download through the VPN of the university of Valencia”. This is not a very scientific reason... Did you contact the authors to ask them for their paper?
Answer: Of the 4 articles that could not be accessed, three were eliminated because they did not meet the inclusion criteria. The other article could not be accessed after having tried by different means.
Comment 3: Given that these are studies on aggregated data, at the level of states or countries, there should be a paragraph with the limitations of this kind of studies.
Answer: The limitations of the studies have been included in the tables and in the discussion section. Among these limitations two are important, on the one hand the difference in legalization laws between states and on the other hand, the source of data collection.
Comment 4: It seems that my previous request on commenting in the text only the significative results was not followed : line 180 you mention that 2 papers found an increase in so and so, but we can see in table that one of the 2 (ref 33) has confidence intervals that include zero : they are not significative…If they are not significative, this should at the very least be said in the text.
Answer: As commented by the reviewer, it has been included that the increase is not significant considering the change in trend after legalization. As it was not significant, the subsequent comment on the percentages of drivers testing positive before and after legalization was removed.
Comment 5: Why did you make table1 separated from table2, since they contain the same studies? It would be more helpful to read all about each study in one big table… instead of 2
The same applies for the pair {table 3 and 4}, the pair : tables 5 and 6.
Answer: Initially everything was in the same table. The tables were modified based on the comments of a previous reviewer.
Comment 6:
https://www.bast.de/Druid/EN/deliverales-list/downloads/Deliverable_2_1_3.html;jsessionid=3AC071FF172D84368E679B3B49DA64F2.live11314?nn=613800
In this DRUID report, in the annex, there are a number of tables reporting different studies. They are first structured by type of objective (estimate prevalence while driving or estimate relationship between THC and risk of road crash), and then there are separated by type of studies : table 5, 6, 7. These are tables that are easy to read (even so there should have been more information in each of them). These should be used as a goal for clear tables
The content of each of your tables should much more structured.
It would help if the tables had the same structure as in a paper (introduction /material and methods / results / limitations ):
- general information first (country, years, type of study : before /after or comparison between states or else…), size of samples that are compared, but NOT including results),
- followed by a column with material (as it is now “variables and collected data),
- then by method (statistical analysis and a bit more : what is compared? Or what is estimated?),
- then by results : many more results should be here, rather than in the text (so that it is easier to compare with other studies in the table.
- column limitations : why do we find this “not specified” sometimes? limitations should also include the limitations YOU think there are.
For instance in table 1, study 33: why do you give general frequencies, and “results” in the WHOLE sample? It would be more useful to have the size of the study group before legalization, and the size of the study group after legalization. The results for each of the 2 (or more) study groups should be given in the results column (we don’t need gender and proportion of THC+ for the WHOLE sample : they are not useful for the objective of the paper). Please only provide results that are useful for the purpose of your paper
Please be 1) precise, 2) more specific, 3) more informative
For each line in the table, we need to understand, what was done (what was compared to what, and so on). It should be a good summary of the study: we should not have to go back to the text or to the article itself to understand what was done.
Answer: The sample document sent by the reviewer has been taken into account. In this sense, the Study group column has been included. In addition, sample characteristics have been eliminated and the column has been renamed Sample size, focusing specifically on the specific groups according to the objective of this review.
Comment 7: The distinction between before/after studies and comparison between states in not clear enough. It could help to make sub-tables so that the columns better fit the studies, and so that it is easier to compare them. This of course applies to medical, and/or recreational cannabis studies.
Answer: Initially, priority was given to separating the studies according to the type of marijuana that had been legalized. The authors consider that there is a clear difference between the legalization of medicinal and recreational marijuana according to the uses, benefits and costs for health according to the scientific evidence in this regard. On the other hand, the text specifies the studies that have focused on the moment, pre- and post-legalization, and between states that have legalized it and those that have not.
Comment 8: Please provide a table summarizing the analysis of quality that you did on the selected papers. Please explain why you only use 10 indicators out of 18.
Answer: We selected those that appear to a large extent in the empirical articles and that are of greater relevance. These criteria are included in the method section. Other criteria were not selected because they are not necessarily common to all the papers, such as Funding or Ethical Considerations. The authors of the review considered that selecting 10 of the 18 criteria was sufficient to determine the quality of the papers included in the present study. This same criterion has been used in other articles published by the authors.
Comment 9: The English version needs to be checked by a native English speaker, or at least by someone who is fluent in English. Please make shorter sentences.
Answer: The manuscript has been reviewed by a native English speaker.
Comentario 10: The comments of the other reviewer of the first version of the manuscript do still apply.
Answer: Previous changes requested by other reviewers, such as making separate tables for clarity and formatting, have been included.